# Strategies to Enhance Drug Absorption via Nasal and Pulmonary Routes

**DOI:** 10.3390/pharmaceutics11030113

**Published:** 2019-03-11

**Authors:** Maliheh Ghadiri, Paul M. Young, Daniela Traini

**Affiliations:** Respiratory Technology, Woolcock Institute of Medical Research and Discipline of Pharmacology, Faculty of Medicine and Health, The University of Sydney, Camperdown, NSW 2006, Australia; paul.young@sydney.edu.au (P.M.Y.); daniela.traini@sydney.edu.au (D.T.)

**Keywords:** nasal, pulmonary, drug administration, absorption enhancers, nanoparticle, and liposome

## Abstract

New therapeutic agents such as proteins, peptides, and nucleic acid-based agents are being developed every year, making it vital to find a non-invasive route such as nasal or pulmonary for their administration. However, a major concern for some of these newly developed therapeutic agents is their poor absorption. Therefore, absorption enhancers have been investigated to address this major administration problem. This paper describes the basic concepts of transmucosal administration of drugs, and in particular the use of the pulmonary or nasal routes for administration of drugs with poor absorption. Strategies for the exploitation of absorption enhancers for the improvement of pulmonary or nasal administration are discussed, including use of surfactants, cyclodextrins, protease inhibitors, and tight junction modulators, as well as application of carriers such as liposomes and nanoparticles.

## 1. Background

Absorption enhancers are functional excipients included in formulations to improve the absorption of drugs across biological barriers. They have been investigated for many years, particularly to enhance the efficacy of peptides, proteins, and other pharmacologically active compounds that have poor barrier permeability [1]. The ideal absorption enhancer should be one that protects biological agents against enzymatic degradation and causes a rapid opening of the relevant barrier while enhancing absorption transiently.

As a portal for non-invasive delivery, nasal and pulmonary administration has several advantages over traditional oral medication or injection. Nasal and pulmonary delivery are non-invasive routes of administration that target the delivered dose directly to the site of drug action [2,3]. Moreover, drug delivery to the respiratory area can also be used for systemic delivery of peptides and proteins due to the large surface area for drug absorption. Pulmonary and nasal administration bypasses first-pass metabolism that is observed in oral administration and the lung and nasal cavity have a low drug metabolizing environment [4]. Despite all these advantages, there is a significant challenge to enhance the absorption of the active agent via these routes. In nasal and pulmonary administration, absorption enhancers have been investigated over the last two decades, to increase the rate of absorption by targeting different mechanisms [5,6]. These mechanisms are either to improve the permeation of materials across the epithelial barrier via intracellular or paracellular mechanisms (Figure 1) or to enhance stability and mucus solubility of the drugs regionally. However, to date, no safe absorption enhancer for pulmonary administration of drugs has translated into commercial products. Their use has generated safety concerns due to potential irreversible alteration of the epithelial cell membrane, which could potentially make the lung susceptible to the entry of exogenous allergens. While there are no absorption enhancers for pulmonary drug administration on the market, quite a few appear to be at the threshold of becoming products for nasal administration.

This review critically assesses advances in the field of absorption enhancers for nasal and pulmonary drug administration. The various agents used to increase the absorption of poorly permeable drugs, their mechanisms of action, safety, and effectiveness are also presented.

## 2. The Barriers

The first requirement for drug absorption is for the active pharmaceutical ingredient (API, or drug) to reach the absorbing barrier. For the respiratory system, the drug has to be deposited on the luminal surface of the epithelial membrane and be absorbed before being cleared or degraded. Furthermore, adequate absorption of the API may also require controlling its release profile as it passes through multiple biological barriers. These barriers include: (1) the surfactant layer, mucus layer, (2) epithelial layer, (3) interstitium and basement membrane, and (4) capillary endothelium (Figure 2).

The mucus layer is the first barrier. The deposited drug needs to be dissolved or transverse the mucus layer before degradation due to the enzymatic activity or clearance by mucociliary activity. Because the ciliary action is relatively fast in removing the drug from the absorption site, permeation enhancers must act rapidly to increase bioavailability. The understanding of mucus layer thickness and clearance rate is important to develop drug delivery strategies to overcome mucosal clearance mechanisms. In the lung, the thickness of the luminal mucus gel has been reported to be ~5–10 μm [7]. The underlying, less viscoelastic sol layer, also known as the periciliary liquid, covers the cilia and has an additional 5–10 μm thickness [7]. However, other studies based on confocal fluorescence microscopy suggest that airway mucus may range in thickness from 5 to 55 μm [8,9]. The nasal tract has a thin mucus layer which is readily accessible and considered highly permeable compared to other mucosal surfaces [10]. In the nasal tract, the ciliary motion transports mucus with the flow rate of about 5 mm per minute, and the mucus layer is renewed approximately every 20 min [11,12]. Similarly, the luminal gel layer of respiratory tract mucus is replaced every 10 to 20 min, [11]. While, the sol layer of respiratory mucus has a slower rate of clearance than the more solid-like luminal gel layer. Drug absorption via the nose is dependent on drug clearance from the nasal cavity, which is determined by nasal mucociliary clearance (MC). In a study by Inoue et al., the effect of nasal MC on in vivo absorption of norfloxacin after intranasal administration to rats was estimated quantitatively. This study resulted in a model to precisely estimate nasal drug absorption [13].

The second barrier is the epithelial cell membrane. It is comprised of a layer of pseudostratified columnar cells interconnected via tight junctions. Most drugs are primarily absorbed via transcellular diffusion, permeating through the epithelial cell membrane. Small hydrophobic molecules can partition across biological membranes via a concentration gradient. Hydrophilic molecules generally require some sort of selective transport system to cross the lipid bilayer. Large and polar drugs may be absorbed by a paracellular mechanism, and the tight junction structure represents the barrier to paracellular absorption.

Once a drug molecule has passed through to the basolateral side of the epithelium, the next barrier is the capillary endothelium for absorption into the blood. While this is not critical for locally acting drugs, it is important for systemically targeting APIs.

Strategies used to overcome these barriers to absorption include: A. Preventing degradation/metabolism; B. Enhancing barrier permeability via transient opening of tight junctions, C. Disruption of lipid bilayer packing/complexation/carrier/ion pairing; and D. Enhancing resident time/slowing down mucociliary clearance (Figure 3).

In the following sections, absorption enhancers utilized to overcome these barriers in nasal and pulmonary routes of administration are discussed.

## 3. Absorption Enhancers Investigated for Nasal and Pulmonary Drug Administration

The most common permeation enhancers investigated for nasal and pulmonary drug administration and their classification are listed in Table 1. Most of these agents have been divided into one of the following major classifications: (I) surfactants, (II) cyclodextrins, (III) protease inhibitors, (IV) cationic polymers, and (V) tight junction modulators. Each of these classifications are discussed in greater detail below.

### 3.1. Surfactants

Surface active agents, or surfactants, are amphiphilic molecules possessing both lipophilic and hydrophilic residues. Surfactants have various applications in pulmonary drug administration, due to their high interfacial activity; one of which is as an absorption enhancer [30]. Surfactants can enhance absorption with more than one mechanism; these include perturbing the cell membrane by leaching of membrane proteins, opening of tight junctions, or preventing enzymatic degradation of the drugs [31]. These surfactants are mainly used and studied in oral drug administration; however, there have been a few studies looking at their application in nasal and pulmonary drug delivery. Surfactants used as absorption enhancers can be classified as; (A) phospholipids [32], (B) bile salts, (such as sodium taurocholate etc.) [33], (C) non-ionic surfactants [31], (D) salt of fatty acids [20], and (E) alkyl glycosides (e.g., tetradecylmaltoside, *N*-lauryl-*b*-*d*-maltopyranoside etc.) [34].
(a)**Phospholipids**: Natural pulmonary surfactant is a complex mixture of phospholipids (90%) and proteins (10%). The main function of this surfactant is to reduce the surface tension at the alveolar air–liquid interface of the lungs to avoid alveolar collapse [35]. It has also been shown that phospholipids can enhance the absorption of active agents in the lung [36]. A review by Wauthoz extensively discussed their role in pulmonary drug administration [36]. For example, Dipalmitoylphosphatidylcholine (DPPC) is a main component of lung surfactant, representing 40% by weight [37]. DPPC has been used as a lung absorption enhancer in several studies [23,38,39]; for instance, it was used to optimize the absorption of parathyroid hormone 1-34 (PTH) from the lungs into the bloodstream [38]. Other lung surfactants, including phosphatidyl cholines (35%), phosphatidylglycerol (10%), phosphatidylinositol (2%), phosphatidylethanolamine (3%), sphingomyelin (2.5%), and neutral lipid (3%) have also been used as absorption enhancers. These natural pulmonary surfactants (PS) and their artificial substitute phospholipid hexadecanol tyloxapol (PHT) have been tested as absorption enhancers for promoting recombinant human insulin (Rh-ins) absorption in vivo from the lung in a diabetic rat model [40]. In another study, the same phospholipids were tested in vitro on Calu-3 ALI (air-liquid culture) model [41] to further investigate their absorption potential. This in vitro study demonstrated an enhanced permeation of Rh-ins and fluorescein isothiocyanate-labelled dextran (FD-4) (4000 Dalton molecular weight) through the cell layer. Hence, PS demonstrated a greater absorption enhancing effect than that of PHT. However, they could not identify the underlying mechanism of enhanced absorption. It was suggested that PS and PHT may interact directly with the tight junctions and increase the absorption via the paracellular pathway.(b)**Bile salts**; one of the primary roles of bile salts and their derivatives in drug delivery is their ability to enhance absorption [42]. For pulmonary drug delivery applications, salts of cholate, deoxycholate, glycocholate, glycodeoxycholate, taurocholate, and taurodeoxycholate [43] have been tested as absorption enhancers. Sodium taurocholate is one of the most-used bile salts to increase bioavailability of proteins, especially insulin [33] via the pulmonary route. The ranking of enhancement by bile salts for insulin has been reported to be sodium deoxycholate > sodium cholate > sodium glycocholate > sodium glycodeoxycholate (GDCA) > sodium taurodeoxycholate [33]. Even though bile salts and derivatives have shown potential as absorption enhancers, their toxicity on the epithelial surface is a main challenge in clinical applications. In a recent study the effect of inhaled bile salts on lung surfactant function as absorption enhancers was investigated in two in vitro models and then correlated to in vivo lung effects [44]. This study demonstrated that bile salts in vitro disrupted surfactant function and in vivo induced pulmonary irritation. Therefore, even though the bile salts did not affect the barrier integrity or viability of human airway epithelial cells at the tested doses, they have shown toxicity to some extent.(c)**Fatty acids**; Fatty acids, polyunsaturated fatty acids (PUFA), and their salts have also been investigated as absorption enhancers via the nasal and pulmonary route [21]. They have shown a tight junction modulatory effect to some extent, and enhanced permeation of drugs through the epithelial cell barrier. Although the exact mechanism is still unknown, previous studies have suggested that they may alter the membrane’s permeability, increasing fluidity of the membrane or through Ca^2+^ dependent tight junction mechanisms [45]. Their potential as an absorption enhancer has been demonstrated in both in vitro [20] and in vivo studies [16]. For example, the effects of arachidonic acid as an absorption enhancer combined with amino acid Taurine enhanced absorption of fluorescein isothiocyanate 4000 (FD-4) via the pulmonary route [46]. Among the fatty acids, medium chain fatty acids such as capric acid and lauric acid have been studied extensively as absorption enhancers due to their safety and effectiveness [45,47]. The suggested mechanism for capric acid (sodium caprate) is most likely by activation of phospholipase-C and increase of intracellular calcium levels, resulting in contraction of actin microfilaments and dilation of the tight junctions [45].(d)**Non-ionic surfactants**—Non-ionic surfactants, consisting of a hydrophilic head group and a hydrophobic tail, carry no charge and are relatively non-toxic [48]. Poloxamer 188, a non-ionic surfactant, has been widely studied as intranasal drug delivery system [49]. in vitro and in vivo studies demonstrated that poloxamer 188 played a key role in promoting intranasal absorption of both isosorbide dinitrate [49] and sumatriptan succinate [50] in rats. Incorporation of poloxamer 188 was reported to be able to influence the elasticity of nano-cubic vehicles for intranasal delivery [51]. Other non-ionic surfactants such as cremophor EL, laurate sucrose ester (SE), and sucrose cocoate have also shown absorption enhancement properties via nasal administration. SE has shown an efficient absorption-enhancing effect of poorly permeable drugs [31], furthermore, intranasal administration of an insulin formulation containing 0.5% sucrose cocoate showed a rapid and significant increase in plasma insulin level, with a concomitant decrease in blood glucose level [52]. Alkylglycosides (AGs) are a type of non-ionic surfactant class with groups such as maltose, sucrose or monosaccharides (e.g., glucose) attached to alkyl chains of variable length. Tetradecylmaltoside and *N*-lauryl-*b*-*d*-maltopyranoside are the most commonly used AGs. They have shown effective nasal absorption enhancement properties at extremely low concentrations. Pillion and his colleagues showed that AGs could be used effectively to enhance nasal absorption of insulin, calcitonin, and glucagon [53]. They synthesized a series of new glycosides with extended alkyl side-chains (C13–16) linked to maltose or sucrose and tested their efficacy as a penetration enhancer for delivery of nasal insulin in anesthetized rats [54]. Of the AGs tested, tetradecyl maltoside (TDM), a 14-carbon alkyl chain attached to a maltose ring, has been shown to be the most efficacious in enhancing nasal insulin absorption. The effects of TDM on the respiratory epithelium were shown to be reversible, with the epithelium reversing to its normal physiological barrier function 120 min after exposure to these agents. The molecular mechanism involved in these absorption-enhancing effects in vivo is unclear. It has been suggested that AGs have a direct effect on the epithelium layer, probably via the para cellular pathway [55]. Although AGs have shown absorption enhancing effect, they exhibit significant toxicity towards airway epithelial cells (Calu-3 cells), probably from a membrane-damaging effect [56].(e)**Bio-surfactants**—Biosurfactants are surface-active substances synthesised by living cells such as bacteria, fungi, and yeast. Bio-surfactants are generally non-toxic, environmentally benign, and biodegradable. Biosurfactants have been investigated as drug absorption enhancers previously [57]. One of the most well characterized classes of biosurfactants are rhamnolipids. The effect of rhamnolipids on the epithelial permeability of FD-4 and FD-10 across Caco-2 and Calu-3 monolayers has been reported [58]. It was shown that rhamnolipids increased permeation of FD-4 and FD-10 across both cell lines at a safe concentration with a dose-dependent effect.(f)**Animal derived surfactants**—The animal-derived surfactant poractant alfa (Curosurf®) was used to deliver polymyxin E and gentamicin to the lung in a neonatal rabbit model [59]. In this study, polymyxin E was mixed with poractant alfa and administered to a near-term rabbit. This mixture increased bactericidal effect of the antibiotics against Pseudomonas aeruginosa in vivo. This may be due to more efficient spreading mediated by interactions between drugs and surfactant.

### 3.2. Enzyme Inhibitors

Airway surface liquid and mucus contains a high number of enzymes including proteases, and nucleases that may degrade active agents before they are absorbed [60]. Among these enzymes, serine proteases and aminopeptidases constitute the majority of degrading enzymes present in the lung. Given the high number of enzymes in the lung, these may metabolize respiratory drugs before they reach the absorbing membrane. Peptide, proteins, and nucleic acid-based drugs are especially vulnerable to metabolism. Therefore, a strategy to protect drugs against enzymatic degradation within the lung and nasal cavity may be necessary in some cases. Some of the protease inhibitors studied over the last decade in pulmonary/nasal drug administration as absorption enhancers are: nafamostat mesilate [61], aprotinin [62], bacitracin [63], soybean trypsin inhibitor [63], phosphoramidon [15], leupeptin [64], and bestatin [65]. In one study, aprotinin, bacitracin, and soybean trypsin were used as protease inhibitors in combination with other absorption enhancers (sodium glycocholate, linoleic acid-surfactant mixed micelles, and *N*-lauryl-β-d-maltopyranoside) and tested on rat lung for the absorption of insulin and Asu (1,7)) eel-calcitonin (ECT) [62,63]. During the aforementioned study, rats received insulin with protease inhibitor via intra tracheal administration. The absorption of insulin from the lung was evaluated by their hypoglycemic and hypocalcemic response when used with these additives [62,63]. In the presence of protease inhibitors, the plasma concentration of glucose reached a minimum of 24.0–66.7% of baseline within 90 min of solution administration. It was also found that bacitracin (20 mM) was more effective at enhancing the pulmonary absorption of insulin than aprotinin and soybean trypsin inhibitor. In the same study, the effects of these protease inhibitors on the stability of insulin in rat lung homogenate were investigated, to elucidate the enhancing mechanisms of these protease inhibitors. All protease inhibitors were effective in reducing insulin degradation and these findings suggest that combination of absorption enhancers and protease inhibitors would be a useful approach for improving the pulmonary absorption of biologically active drugs. Studies have shown that when nafamostat mesilate, which strongly inhibits a variety of proteases, such as trypsin, plasmin, and kallikaren, was co-administered with insulin in the lung, the relative bioavailability of insulin was approximately twice that obtained when the peptide was administered alone [66].

During chronic lung inflammation and infection proteases such as neutrophil elastase (NE), a neutrophil-specific serine protease against P. aeruginosa, will be released into the lung lumen to fight pathogens involved in lung infections [67]. Excessive accumulation of NE in pulmonary fluids and tissues of patients with chronic lung infection is thought to reduce the absorption of inhaled drugs. Therefore, neutrophil elastase inhibitors (NEIs) have shown potential as absorption enhancers by protecting drug moieties from degradation by neutrophil elastase [68]. The potential use of NEIs, such as peptide chloromethyl ketones or reversible peptide aldehydes, tripeptide ketones, modified NE-specific β-lactams, or peptide boronic acids have been largely replaced by the development of EPI-HNE-4, a rapid acting and potent NE inhibitor [69] which can potentially be nebulized to CF patients [70]. However, clear clinical efficacy of this NE inhibitor remains to be demonstrated.

### 3.3. Cationic Polymers as Absorption Enhancers

Polymeric systems with positive charges or modified with cationic entities, incorporated on their backbone and/or side chains, are considered cationic polymers [71]. Cationic polymers have the potential to enhance absorption of macromolecules [72]. Cationated gelatins [72], cationated pullulans, poly-l-arginine, polyethylenamine (PEI), chitosan, and its derivatives are types of cationic polymers. Cationic polymers interact with the mucosal barriers and enhance the absorption of water-soluble macromolecules via tight junctions modification. For example, in the case of insulin with negative charges in neutral solutions, interaction between the cationic polymers and insulin is important to promote effective insulin absorption. An appropriate interaction can help insulin to access the cell surface; however, strong interaction can inhibit insulin absorption. PEI is a cationic and highly water-soluble polymer that has shown potential as a carrier for nasal drug administration. It was also demonstrated that the degree of positive charge was linearly correlated with absorption enhancing effect of PEI, suggesting that positive charge of PEI might be related to its absorption mechanisms for enhancing pulmonary absorption of insulin in rats [73]. Spermined dextran (SD), a cationic polymer, has been studied as absorption enhancer for pulmonary application of peptide drugs [74]. Its enhancing effects on the absorption of insulin and permeation of FD-4 through Calu-3 cells increased with an increase in the molecular weight of SD, over the range 10–70 kDa. The mechanism of action of SD is not fully understood yet, but it is hypothesized that the molecule may interact directly with the luminal surface of the mucus barriers via an ion-ion interaction, inducing the opening of tight junctions, resulting in intercellular permeation of water soluble drugs [74]. Chitosan and its derivatives are excellent examples of cationic polyelectrolytes. They have been extensively used to develop mucoadhesive polymers [25,75] and have favorable characteristics such as biocompatibility, biodegradation, and low toxicity. These aforementioned characteristics make them suitable as a pharmaceutical excipient [76]. Chitosan interacts electrostatically with the negatively charged mucin chains thereby demonstrating mucoadhesive properties. This mucoadhesion prolongs the residence time of the drug and thus enhances API absorption [77]. However, Chitosan derivatives are poorly soluble in water at physiological pH, limiting their application. Chitosan exhibits excellent mucoadhesive properties when dissolved in neutral or alkaline medium so to overcome solubility issues, different derivatives have been synthesized [78]. Chitosan oligomers [79], for example, have relatively high solubility in water compared to conventional chitosan and have been tested for their absorption enhancing potential via lung [25]. For example, the pulmonary absorption of interferon-α has been shown to be effective when using chitosan oligomers [25]. Of these chitosan oligomers, 0.5% *w*/*v* chitosan hexamer appeared to be more effective in enhancing the pulmonary absorption of IFN than other oligomers at the same concentration, and the AUC value of IFN with chitosan hexamer increased 2.6-fold when compared to control. In a recent study, O-palmitoyl chitosan, synthesized from chitosan and palmitoyl chloride, demonstrated improved mucoadhesive and absorption enhancing properties [80]. In addition, bioadhesive properties of chitosan may be useful in enhancing drug absorption following inhalation [81].

Sperminated pullulans (SP) have been shown to enhance pulmonary absorption of insulin in rats, with their enhancing effects correlated to the amino group content and their molecular weight [82]. SP acted as an enhancer for insulin absorption when a 0.1% *w*/*v* solution was applied with insulin simultaneously in vivo. Ikada et al. studied the use of negatively and positively charged gelatin microspheres for pulmonary administration of salmon calcitonin in rats [83]. The pharmacological effect after administration of salmon calcitonin in positively charged gelatin microspheres was significantly higher than that in negatively charged gelatin microspheres. Additionally, administration of salmon calcitonin in positively charged gelatin microspheres with smaller particle sizes led to a higher pharmacological effect. The pharmacological effect after pulmonary administration of salmon calcitonin in positively charged gelatin microspheres with particle sizes of 3.4 and 11.2 µm was approximately 50% [83].

Polyamines have also been tested for their absorption enhancing properties [84]. The polyamines spermidine and spermine are commonly found in all mammalian cells [84]. They are essential for the maintenance of cell growth in many tissues. Specifically, for the lungs, He et al. showed that polyamines, particularly spermine and spermidine, can effectively improve the pulmonary absorption of insulin and other water soluble macromolecules without any membrane damage of the lung tissues in rats [84]. It was suggested that the absorption-enhancing mechanism of spermine partly includes opening of the epithelial tight junctions. Sperminated dextrans have also been studied as absorption enhancers with different average molecular weights (MWs; 10, 40, and 70 kDa) and numbers of amino groups, prepared as cationized polymers [74]. Sperminated dextrans increased pulmonary absorption of insulin in rats and also permeation of FD-4 through Calu-3 cell monolayers in vitro [74].

### 3.4. Tight Junction Modulators

The intercellular spaces between adjacent epithelial cells are sealed by tight junctions (TJs). Modulation of TJs is an effective strategy for drug absorption via the paracellular pathway. The paracellular transport is not suitable for the transport of large macromolecules and is generally restricted to the compounds of molecular radii less than 11 Å. Hydrophilic drugs with low molecular weight, peptides, and proteins often have poor bioavailability. However, it has been shown that some peptide drugs, such as octreotide, desmopressin, and thyrotropin-releasing hormone are absorbed by this route in which tight junctions play a fundamental role [85]. Tight junction modulators effective on TJ proteins such as Claudin and ZO are extremely potent in opening these tight junctions, 400 fold stronger than other agents [86]. Until now they have mainly been tested on intestinal and dermal [87] tight junction barriers [88,89] and on the blood–brain barrier [90]. Of these modulators, two have been tested for pulmonary drug administration.

**(A)** Clostridium perfringens enterotoxin (CPE)—The C-terminal fragment of CPE (C-CPE) is known to modulate the barrier function of claudin [28]. Claudin is one of the key structural and functional components of the TJ-seal (70). Therefore, it has been suggested that claudin may be a potential target for paracellular API delivery. C-CPE is a potent absorption-enhancer and this enhancing activity is greater than clinically used enhancers. The main problem with tight junction modulators is their toxicity [91]; therefore, many variants of CPE have been synthesized to decrease toxicity [86]. In a study on nasal and pulmonary absorption of human parathyroid hormone hPTH in rats, C-CPE was used as an absorption enhancer. It was instilled into each nostril and after 4 h hPTH was delivered intranasally. A Micro-sprayer was used to spray C-CPE into rats’ lungs, then after 4 h, hPTH was administered. This study showed C-CPE, and enhanced nasal but not pulmonary absorption of hPTH [89].

**(B)** Zonula occludens toxin (ZOT) are another tight junction modulator. Zot is a protein of Vibrio cholera, and zonulin is the Zot analogue that governs the permeability of intercellular TJs [92]. Zot and Zot derivatives are reversible TJ openers that enhance the delivery of drugs through the paracellular route. The active domain of zonula occludens toxin (ZOT) is called AT-1002 [93]. AT-1002, a hexamer peptide, induced tight junction disassembly in the epithelial layer in the trachea [29]. In this study, in vivo intratracheal administration of salmon calcitonin in rats with 1 mg of AT-1002 resulted in a 5.2-fold increase in absorption of calcitonin over the control group [29].

## 4. Other Strategies to Enhance Absorption

### 4.1. Nanoparticles (NPs)

Nanoparticles (NPs) have been used as drug carriers for overcoming mucociliary clearance and to avoid phagocytosis by alveolar macrophages, hence increasing the absorption of drugs in respiratory system [94]. It was shown that alveolar macrophages are less efficient to phagocytose ultrafine particles compared to larger particles [95] and NPs overcome the mucus barrier and achieve longer retention time at the cell surface, effectively penetrating the mucus layer and accumulating on the epithelial surface. In order to penetrate mucus, nanoparticles must avoid adhesion to mucin fibres and be small enough to avoid significant steric inhibition by the dense fibre mesh [96]. In a study by Schneider, variously sized, polystyrene-based muco-penetrating particles (MPP) were synthesized and their absorption in the lungs following inhalation investigated. They demonstrated that MPP as large as 300 nm exhibited uniform distribution and markedly enhanced retention in the lung [96]. NPs can also improve nose-to-brain drug delivery, since they are able to protect encapsulated drug from biological and/or chemical degradation, and from extracellular transport by P-gp efflux proteins [97]. Retention of bioadhesive NPs on the mucosal surface of the nasal cavity, as well as their ability of transiently opening of the tight junctions of the mucosal epithelium, should contribute to enhanced nasal absorption [98].

### 4.2. Liposomes

An approach to enhance mucosal penetration is to encapsulate drugs within liposomes. Liposomes consist of one or more phospholipid bilayers and can incorporate hydrophilic substances within their inner cavity and hydrophobic substances within the lipid bilayer. Liposomes and phospholipids have been investigated for the systemic absorption of proteins after intra tracheal administration [99]. It is suggested that the mechanism of absorption is driven by phospholipids in the liposomes being infused with the rapidly recycling lung surfactant that leads to enhanced uptake of the protein molecule into the systemic circulation [99]. Liposomes are also known to promote an increase in drug retention time and reduce the toxicity of drugs after administration [23]. Several factors influence drug release and absorption of drugs loaded into liposomes such as the composition of lipids and the size of the liposomes used [100]. The presence of cholesterol and phospholipids with saturated hydrocarbon chains have also been shown to increase drug residence time within the lung [23]. A number of nebulizable liposome formulations have reached clinical trial phase. For example, Arikace^®^ (liposomal amikacin) and Pulmaquin^®^ (liposomal ciprofloxacin) are antibacterial formulations currently in advanced stages of development.

Glycerosomes are vesicles composed of phospholipids, glycerol, and water, and have been used as novel vesicular carriers for trans-mucosal drug delivery. Glycerosomes are made from a diverse range of phospholipids and a high percent of glycerol (20–40%, *v*/*v*). They are flexible vesicular carriers, where the glycerol component alters the vesicle membrane fluidity, and are constituted by different substances such as cholesterol that enhance the lipidic bilayer stability. They also may contain basic or acidic lipid molecules which adjust the electrical charge of vesicular surfaces and decrease liposome aggregation. Glycerosomes can be prepared by the same common techniques that are used for the preparation of conventional liposomes.

As an example, curcumin was incorporated in glycerosomes for delivery into the lungs. In this study, curcumin was loaded into glycerosomes which were then combined with two polymers, sodium hyaluronate and trimethyl chitosan, to form polymer-glycerosomes [101]. The study showed that nebulized curcumin vesicles were able to protect in vitro A549 cells stressed with hydrogen peroxide, restoring healthy conditions, not only by directly scavenging free radicals but also by indirectly inhibiting the production of cytokine IL6 and IL8. Also, in vivo results in rats showed the high capacity of these glycerosomes to favor the curcumin accumulation in the lungs, confirming their potential use as a pulmonary drug delivery system. In another study, Rifampicin loaded glycerosomes, vesicles composed of phospholipids, glycerol, and water, were combined with trimethyl chitosan chloride (TMC) to prepare TMC-glycerosomes or with sodium hyaluronate (HY) to obtain HY-glycerosomes [102]. These new hybrid nanovesicles were tested as carriers for pulmonary delivery of rifampicin. Rifampicin nanoincorporation in vesicles reduced the in vitro drug toxicity on A549 cells, as well as increasing its efficacy against Staphylococcus aureus. Finally, the in vivo biodistribution and accumulation, evaluated after intra-tracheal administration to rats, confirmed the improvement of rifampicin accumulation in lung.

### 4.3. Dendrimers

Dendrimers possess high water solubility, and as highly efficient absorption promoters, they can easily penetrate through barriers. In addition, they can be used as carriers for different routes of drug administration [103,104]. Dendrimers have been studied intensively as a drug carrier in delivery systems [103,104]. As a novel class of artificial macromolecules, polyamidoamine (PAMAM) dendrimers have shown excellent performance in drug delivery systems due to their unique physical and chemical properties [105]. PAMAM dendrimers with generation 0 to generation 3 (G0–G3) and concentrations (0.1–1.0%) were tested in terms of their capacity to enhance pulmonary absorption of macromolecules [105]. The results showed that treatment with a 0.1% G3 PAMAM dendrimer could increase the secretion of organic cation transporters (OCTs), OCT1, OCT2, and OCT3, which might be related to the absorption-enhancing mechanisms of the pulmonary absorption of the macromolecule.

### 4.4. Exosomes

Exosomes are a subgroup of 30–100 nm size extracellular vesicles (EVs) secreted by cells into the extracellular environment. EVs have the distinct advantage that their membranes are structurally similar to the cell membrane. This means that EVs lipid composition, fluidity, protein membranes, and other fusogenic proteins are similar to what is found in cell membranes. Because of this unique property, cellular uptake of EVs surpasses that of more traditional carriers, such as liposomes or nanoparticles. Exosomes can be used as a biological nano-platform for enhanced drug delivery [106]. Their advantages include their small size for penetration into deep tissues, slightly negative zeta potential for long circulation, and deformable cytoskeleton, as well as their similarity to cell membranes [107]. In addition, some exosomes also exhibit an increased capacity to escape degradation or clearance by the immune system [108]. Exosomes have been developed as drug delivery vehicles for a variety of drugs [109,110]. Tumor-derived exosomes are of great significance for guiding the targeted therapy of lung cancer, and exosomes themselves can be a target for treatment. For example, GW4869, a neutral sphingomyelase inhibitor (regulates ceramide biosynthesis, promotes exosomes inward budding), tested in mice, demonstrated inhibition of exosomes production with reduced metastasis in lung cancer [111]. In another study, exosomes derived from curcumin-treated cells alleviated oxidative stress, tight junctions (ZO-1, claudin, occludin), adherent junction (VE-cadherin) proteins, and endothelial cell layer permeability [112]. A nano-formulation consisting of exosomes loaded with paclitaxel (PTX), a commonly used chemotherapeutic agent, developed by Batrakova et al. [113], showed efficacy in the treatment of multi-drug resistant cancer cells. Incorporation of PTX into exosomes increased cytotoxicity more than 50 times in drug resistant MDCKMDR1 (Pgp+) cells and showed a potent anticancer effect in the murine Lewis lung carcinoma pulmonary metastases model. It was shown that exosomes loaded with PTX may alter drug intracellular trafficking and bypass the drug efflux system. The potential of intranasally administered exosomes as delivery vehicles for the treatment of neuro inflammatory diseases has also been investigated. After intranasal administration of exosomes loaded with potent antioxidant, considerable amounts of catalase were detected in a Parkinson’s disease (PD) mouse brain model. It provided significant neuroprotective effects in in vitro and in vivo models of PD [114]. Low molecular antioxidant [115], anticancer agents, doxorubicin (Dox) [116], and a model drug rhodamine [117], have also been loaded into exosomes or exosome-like vesicles for nasal drug delivery to enhance absorption and efficacy.

### 4.5. Cell Penetrating Peptides (CPPs)

Improving the translocation of drugs across the plasma membrane will significantly enhance their absorption. Therefore, using cell penetrating peptides (CPPs) to enhance drug penetration has the potential to significantly reduce the quantity of drug to be administered, thus reducing possible side effects. Cell-penetrating peptides are short peptides that enable cellular intake/uptake of various molecular moieties with poor permeability across epithelial and endothelial barriers [118]. Several types of cargoes, for example, proteins, nucleic acid based macromolecules such as siRNA, plasmid DNA, and small drug molecules, can be transported by CPPs to overcome the natural cellular biological barriers [119]. The mechanism of action of CPPs is still a matter of some debate. Some research suggests CPPs could pass through the plasma membrane via an energy-independent pathway, with others claiming the formation of micro-micelles at the membrane [120], or direct translocation through the lipid bilayer [121]. Multiple studies have consolidated the high efficiency of CPP-mediated drug delivery in vitro. For instance, CPPs have been used successfully to deliver macromolecules [122], oligonucleotides [119,123], and peptides [124] across different cellular barriers. Specifically, CPPs have been recently investigated in nasal and pulmonary drug administration [125,126,127], where they have been used as a conjugate to liposomes and nanoparticles to enhance absorption of model DNA [128]. Co-administration of CPPs could improve nose-to-brain drug transport. In Kamei’s study, it was demonstrated that, insulin was transported into the brain when co-administered with amphipathic CPP, and eventually insulin reached the deeper regions of the brain such as the hippocampus in both mice and rat [126]. The immunohistological examination of the hippocampus demonstrated enhanced nose-to-brain delivery of insulin had a partial neuroprotective effect but unexpectedly increased amyloid β plaque deposition. Therefore, CPPs seem to hold great promise as delivery agents for biomacromolecules. However, CPP-mediated delivery is apparently not tissue- or cell-type specific, so for specific targeting purposes additional agents need to be included in the drug delivery system.

### 4.6. Surface Modification

To enhance the absorption of poorly permeable drugs, another strategy is to modify the inhaled particle surface with agents that enhance their absorption. For example, by coating particles with lipids, the disturbance of the particles on the cellular layer can be reduced [129]. Moreover, the lipid-coated particles can be readily enfolded by the surfactant layer to form vesicular structures that can fuse with the cell membrane. Polymer coating of nanoparticles and liposomes is another strategy to enhance the mucosal penetration of particles via nasal/pulmonary administration [130]. For example, spray dried polymer coated liposomes composed of soy phosphatidylcholine and phospholipid dimyristoyl phosphatidylglycerol coated with alginate, chitosan, or trimethyl chitosan, increased penetration of liposomes through the nasal mucosa over uncoated liposomes when delivered as a dry powder [130]. Surface-modified liposomes for pulmonary administration of peptides were also investigated. For example, chitosan oligosaccharide (oligoCS) and polyvinyl alcohol (PVA-R) with a hydrophobic anchor were used as surface modifiers [131]. The effect of surface modified liposomes on potential toxicity via inhalation was evaluated in vitro and in vivo. In vitro studies on alveolar epithelial cells (A549 cells) demonstrated that PVA-R modification reduced interaction with this cell line, whereas oligoCS modification electrostatically enhanced cellular interaction. The therapeutic efficacy of a peptide (elcatonin) after pulmonary administration to rats was significantly enhanced and prolonged for 48 h after separate administration with oligoCS- or PVA-R-modified liposomes. Furthermore, oligoCS-modified liposomes increased residency of liposomes in the lung and had a tight junction opening effect. On the other hand, PVA-R-modified liposomes induced long-term retention of elcatonin in the lung fluid, leading to sustained absorption. Lactoferrin, a natural iron binding protein whose receptor is highly expressed in both respiratory epithelial cells and neurons, was utilized to facilitate the nasal delivery of nucleic based therapeutic agents. For example, a Lactoferrin-modified PEG-*co*-PCL NPs was recently developed to enhance brain delivery of a neuroprotective peptide—NAPVSIPQ following intranasal administration [132]. In another study, rotigotine, dopamine agonist for the treatment of PD, was loaded in lactoferrin (Lf) modified poly(ethylene glycol)–poly(lactic-*co*-glycolic acid) (PEG-PLGA) nanoparticles. Following intranasal administration, brain delivery of rotigotine was more effective with Lf-NPs than with NPs alone. The brain distribution of rotigotine was heterogeneous, with a higher concentration in the striatum, the primary region affected in PD. This strongly suggested that Lf-NPs enable the targeted delivery of rotigotine for the treatment of PD.

### 4.7. Cyclodextrins (CDs)

Cyclodextrins (CDs) are a distinct family of chemical reagents that contain six, seven, or eight monosaccharide units in a cyclized ring with a central cavity that can accommodate other agents [133]. Hydrophobic drug molecules or hydrophobic parts of drugs are introduced into the CD apolar cavities, thereby presenting the potential to modify the properties of the inhaled drug. For example, Shimpi et al. showed that CDs have an effect on absorption of hydrophilic macromolecules through the pulmonary and nasal routes [27]. They have demonstrated many benefits in pulmonary drug administration, such as improvements in aqueous solubility, systemic absorption and bioavailability of drugs. It has also been shown that modified CDs, such as methylated β-cyclodextrin (M-β-CD), dimethyl-β-cyclodextrin (DM-β-CD), and hydroxypropyl-beta-cyclodextrin (HP-β-CD) as its derivatives notably enhance intranasal absorption of drugs [134]. β-cyclodextrins (β-CD) have been broadly studied as an intranasal absorption enhancer [49]. The effect of CDs on the respiratory cell layer permeability was investigated in vitro and shown to be concentration-dependent and variable according to the type of CDs, type of chemical modification, and degree of substitution [135]. It has been shown that CDs, in general, do not cause a decrease in cell viability at concentrations ≤1 mM, whereas differences between various CDs were observed at concentrations ≥2 mM [136,137]. HP-β-CD and natural-CDs are safest in terms of cytotoxicity, while M-β-CD were the least safe for pulmonary administration [138]. Particularly, it has been shown that CDs are more effective in animal studies compared to in vitro studies as absorption enhancers [134].

The mechanism of absorption enhancement with CDs is still unclear; however, there are reports that M-β-CD increase transcellular, as well as paracellular, movement of peptide drugs [135]. CDs may also have a direct disruptive effect on the alveolar epithelial membrane as evidenced by the extraction of membrane lipids and proteins [139]. CDs derivatives can also stimulate transmucosal absorption of peptide drugs. Importantly, M-β-CD, such as DM-β-CD, strongly enhance transmucosal insulin absorption, whereas unmodified-cyclodextrin has little effect on insulin absorption [139]. It is suggested that both DM-β-CD and tetradecyl-beta-maltoside cyclodextrin (TDM-CD) enhance absorption of insulin by different mechanisms. It has been demonstrated that CDs enhance transmucosal absorption of insulin by formation of an inclusion complex, with insulin or by direct action on the membrane [139]. The latter may involve removal of membrane proteins, complexation with different membrane components, or inhibition of proteolytic enzyme activity. However, it has been claimed that TDM-CD may act by opening cell–cell tight junctions. There is a commercial product of CDs available in the market called Kleptose^®^ HPB (hydroxypropylβ-cyclodextrin) which has been suggested as an attractive excipient for nasal and pulmonary drug administration due to its potential in solubilizing drugs, enhancing absorption of drugs, and low toxicity profile.

## 5. Products in Development

Currently, there are few products in development that employ an absorption-enhancing technology for nasal and pulmonary formulations, as seen in Table 2.

## 6. Conclusions

Several technologies for enhancing the absorption of poorly permeable therapeutic biomolecules have progressed from early studies. They have demonstrated permeation enhancement in isolated barrier model, and a number of absorption-enhancing technologies, especially for nasal applications, are now in clinical trials. These absorption enhancers increase systemic absorption of biomolecules as indicated by improved bioavailability or bioactivity, and appear to represent possible alternatives to existing products, which have sub-optimal bioavailability. Still, one of the challenges towards clinical application of these agents is their lack of safety. Understanding the mechanism of absorption enhancement may be very useful toward reducing their side effects and improving their efficacy before these could be considered as future platforms for inhalation formulation of drugs with poor permeability.

## Figures and Tables

**Figure 1 pharmaceutics-11-00113-f001:**
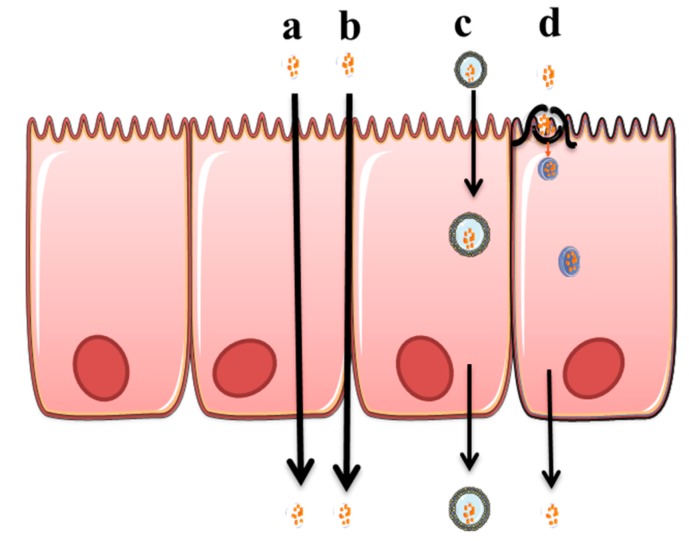
Mechanisms of absorption includes; (a) Transcellular diffusion, (b) Para cellular transport, (c) Vesicle mediated transport, and (d) Carrier mediated transport.

**Figure 2 pharmaceutics-11-00113-f002:**
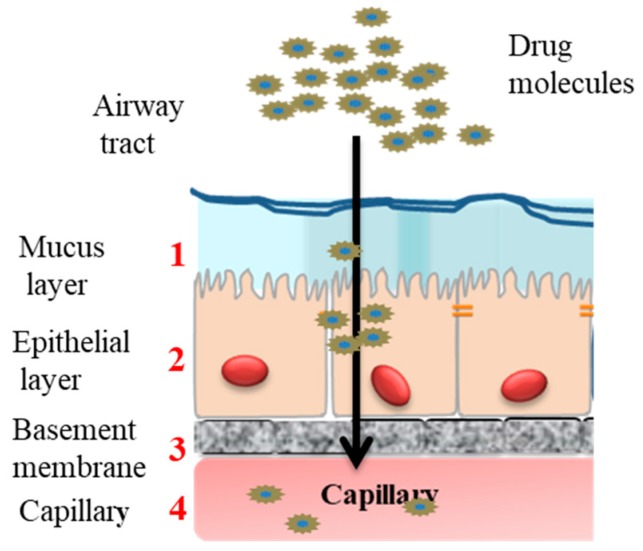
Barriers against drug absorption in pulmonary drug administration including; (1) mucus layer, (2) epithelial layer, (3) interstitium and basement membrane, and (4) capillary endothelium.

**Figure 3 pharmaceutics-11-00113-f003:**
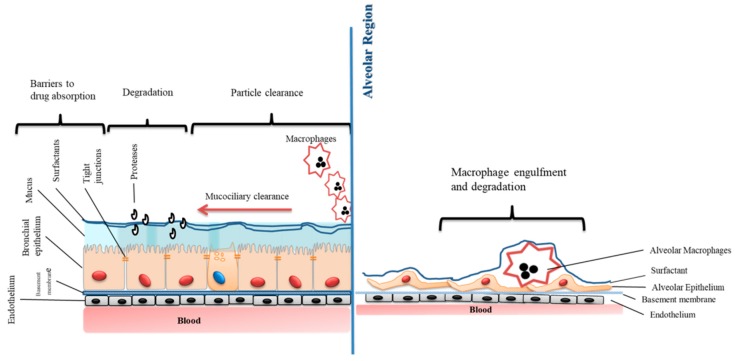
Hurdles for the absorption of drugs via inhalation.

**Table 1 pharmaceutics-11-00113-t001:** Common absorption permeation enhancers- application for nasal and pulmonary administration.

Class	Enhancers	Examples	References
Surfactants	Bile salts	Sodium taurocholate	[14,15]
		Sodium deoxycholate sodium	[16]
		Glycodeoxycholat	[17,18]
Surfactants	Fatty acids and derivatives	Palmitic acid	[19]
		Palmitoleic acid	
		Stearic acid	
		Oleyl alcohol	
		Oleic acid	
		Capric acid	[20,21]
		DHA, EPA	[20]
Surfactants	Phospholipids	Dipalmitoyl phophatidyl choline, soybean lecithin, phosphatidylcholine	[22,23]
Cationic polymers	Polymers	Chitosan and their derivatives	[24,25,26]
Enzyme inhibitors		Human neutrophil elastase inhibitor (ER143)	
Cyclodextrins		Beta-Cyclodextrin	[27]
Tight junction modulators	Claudine modulator	Clostridium perfringens enterotoxin	[28]
	ZO modulator	Zonula occludens toxin (ZOT)	[29]

**Table 2 pharmaceutics-11-00113-t002:** Nasal and Inhalation products in development that employ penetration enhancers.

Technology	Development Stage	Biological Products	Company	Absorption Enhancer Used in the Technology
Cyclopenta Decalactone	Marketed Phase 2	Testosterone (Testim) Nocturia	CPEX Pharmaceuticals Serenity	Surfactant
ChiSys^TM^ PecSys^TM^	Phase 2 Phase 2 Phase 3	Intranasal Apomorphine Intranasal Diazepam Intranasal fentanyl citrate (NasalFent)	Archimedes Pharma Ltd.	Chitosan based delivery
Intravail^TM^	Phase 1 Phase 2 Phase 1	Proteins (IFN-β, EPO) and peptides (PTH, GLP-1), Sumatriptan Naltrexone, Nalmefene	Neurelis, Inc. (Aegis Therapeutics Inc.) Opiant Pharmaceuticals	Cationic polymers-Alkyl saccharide
GelSite^®^ GelVac™ nasal dry powder	Phase 1	Vaccines	Carrington Labs (Delsite Biotech)	Cationic polymers-Poly saccharide
µco™	Phase II Phase 1	Granisetron-zolmitriptan Peptides: (insulin, PTH, FSH, GHRP) Nasal epinephrine formulation	SNBL, Ltd. G2B Pharma Inc.	Polymer-Micro crystalline cellulose (Powder)

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
