# Peer review of "Strategies to Enhance Drug Absorption via Nasal and Pulmonary Routes"

_pharmaceutics, 2019, doi:10.3390/pharmaceutics11030113_

Reviewer 1 Report

Drug absorption enhancement via nasal and pulmonary routes is an important topic.  According to the statement (line 38), “The purpose of this review is a critical assessment of the past 5 years advances in the field of absorption enhancers for nasal and pulmonary delivery.”  The review would be a significant contribution to the field if new advancements would be discussed.  However, among the 116 papers cited in this manuscript, only 38 papers were published in year 2013 and later, and several papers are review articles.  It would be suggested to conduct more literature search and review the most recent publications (published in the past 5 years).  The authors may consider to conduct a critical review of new development in the field while giving a general introduction of the background and earlier publications. 

1.       Section 3: the authors provided extensive background information on different categories of absorption enhancers.  However, new advances in the field were not reviewed or just briefly described.  For example, all references on phospholipids (refs 25-31) discussed were not published within the last 5 years. 

2.       Section 4: the authors reviewed nanoparticles as a strategy to enhance absorption.  However, this section seems to focus on the mucus layer.  It may be more appropriate if this section on the mucus layer could be integrated to section 2 “the barriers”.

3.       A table can be added to summarize the major development in the field.

4.       The previously published review articles on the topic may be cited in the introduction to help readers understand the importance and necessity of this review.

Author Response

Point 1: Section 3: the authors provided extensive background information on different categories of absorption enhancers.  However, new advances in the field were not reviewed or just briefly described.  For example, all references on phospholipids (refs 25-31) discussed were not published within the last 5 years.

Response: The background provided is necessary to understand the mechanism and different types of absorption enhancers, with a focus only on nasal and respiratory drug absorption. Recent review articles have now been added as per the reviewer suggestions to show recent updates on this area.

Point 2: Section 4: the authors reviewed nanoparticles as a strategy to enhance absorption.  However, this section seems to focus on the mucus layer.  It may be more appropriate if this section on the mucus layer could be integrated to section 2 “the barriers”.

Response: This section was amended to include the broader use of nanoparticles as a strategy to enhance absorption.

This section was changed as below:

Nanoparticles (NPs) have also been used as a drug carriers for overcoming mucociliary clearance and alveolar macrophages, hence increase the absorption of drugs via respiratory system. Studies showed that NPs can deposit at the lining fluid and escape both mucociliary clearance and alveolar macrophage.  Alveolar macrophages are less efficient to phagocytose ultrafine particles compared to larger particles. NPs also overcome the mucus barrier and achieve longer retention time at the cell surface, effectively penetrating the mucus layer and accumulating on the epithelial surface. In order to penetrate the mucus, nanoparticles must avoid adhesion to mucin fibres and be small enough to avoid significant steric inhibition by the dense fibre mesh. In a study by Schneider, variously sized, polystyrene-based mucus penetrating particles (MPP) were synthesized and their absorption in the lungs following inhalation investigated. They demonstrated MPP as large as 300 nm exhibited uniform distribution and markedly enhanced retention in the lung.

Point 3: A table can be added to summarize the major development in the field.

Response: Since there have been not been major developments in the last 10 years the only  technologies currently using absorption enhancers are already listed on Table 2.

Point 4: The previously published review articles on the topic may be cited in the introduction to help readers understand the importance and necessity of this review.

Response: As suggested, recent review papers have now been added to the text.

Reviewer 2 Report

  "As a portal for non-invasive delivery, nasal and pulmonary administration have several attractive features."  What are these attractive features?  Please specify.

  Lines 95 - 99:  "Dipalmitoyl phosphatidyl choline (DPPC) has the highest quantity in lung surfactant, representing 40% by weight [27].  ...  Other lung surfactants including; phosphatidyl cholines (35%), phosphatidyl glycerol (10%), phosphatidyl inositol (2%), phospidyl ethanolamine (3%), sphingomyelin (2.5%) and neutral lipid (3%) [25] have been also used as absorption enhancers [26]."   

     a.  "Other lung surfactants including; ..."  --> Other lung surfactant lipid components including ...

     b.  "... ; phosphatidyl cholines (35%), ..."  DPPC has already been mentioned and should not be repeated here.

     c.  " ... phosphatidyl glycerol ..."  --> phosphatidylglycerol   This pertains to the other phospholipids in this list.

 Lines 106 - 107:  FD4   Indicate that this is a fluroscein isothiocyanate-labelled dextran of 4,000 dalton molecular weight.

Line 104:  "Calu 3 ALI"  What is this?

line 134:  "... fluorescein isothiocyanate 4000 (FD-4) ..." --> fluorescein isothiocyanate-labelled dextran    Also, FD4 (line 107) or FD-4?

"Among these proteases, serine proteases and aminopeptidases constitute the majority of the proteases present, with serine, cysteinyl, aspartyl and metalloproteinases the major classes of proteases present in the human lung [57]."    Serine proteases are listed as among the majority and major proteases present.  This is redundant.  Are "cysteinyl, aspartyl and metalloproteases aminopeptidases?  Please clarify.

line 211:  "nucleotide-based drugs"  How do protease inhibitors help in delivery of nucleotide-based drugs?

Author Response

Point 1: "As a portal for non-invasive delivery, nasal and pulmonary administrations have several attractive features."  What are these attractive features?  Please specify.

Response- Attractive feature has been changed to ‘advantages’- Furthermore the background has been modified to include those more clearly.

These non‐invasive routes of administration target the delivered dose directly to the site of drug action. Moreover, drug delivery to the respiratory area is being considered more often for systemic delivery of peptides and proteins. This is due to the respiratory’ large surface area (100 m2), very thin alveolar epithelium (0.2 mm) and low drug metabolizing environment.  Despite all these advantages still there is a significant challenge to enhance the absorption of the active agent.

Point 2- Lines 95 - 99:  "Dipalmitoyl phosphatidyl choline (DPPC) has the highest quantity in lung surfactant, representing 40% by weight [27].  ...  Other lung surfactants including; phosphatidyl cholines (35%), phosphatidyl glycerol (10%), phosphatidyl inositol (2%), phospidyl ethanolamine (3%), sphingomyelin (2.5%) and neutral lipid (3%) [25] have been also used as absorption enhancers [26]."  

     a. “Other lung surfactants including; ...” --> Other lung surfactant lipid components including ...

Response- Amended to “lung surfactants with lipid composition”

     b. “...; phosphatidyl cholines (35%),”  DPPC has already been mentioned and should not be repeated here.

Response- Dipalmitoylphosphatidylcholine or DPPC is a phospholipid consisting of two palmitic acids attached of a phosphatidylcholine head-group and is the major constituent of many pulmonary surfactants. Phosphatidyl choline is a general term for all different types of them.

Point 3- Lines 106 - 107:  FD4   Indicate that this is a fluroscein isothiocyanate-labelled dextran of 4,000 dalton molecular weight.

Response-Amended to fluorescein isothiocyanate-labelled dextran (4,000 Dalton molecular weight)

Point 4- Line 104:  "Calu 3 ALI"  What is this?

Response-Amended to “Calu-3 cells grown on an air-liquid interface (ALI) model”

Point 5- line 134:  "... fluorescein isothiocyanate 4000 (FD-4) ..." --> fluorescein isothiocyanate-labelled dextran    Also, FD4 (line 107) or FD-4?

Response- Amended to FD-4 in the entire manuscript.

Point 6-  "Among these proteases, serine proteases and aminopeptidases constitute the majority of the proteases present, with serine, cysteinyl, aspartyl and metalloproteinases the major classes of proteases present in the human lung [57]."    Serine proteases are listed as among the majority and major proteases present.  This is redundant.  Are "cysteinyl, aspartyl and metalloproteases aminopeptidases?  Please clarify.

Response- To avoid confusion, the sentence has been rephrased to:

“Among these enzymes, serine proteases and aminopeptidases constitute the majority of the degrading enzymes present in the lung”.

Point 7- line 211:  "nucleotide-based drugs"  How do protease inhibitors help in delivery of nucleotide-based drugs?

Response-  The sentence has been amended to “ Given the high number of enzymes like nuclease and proteases present in the lung, these may metabolize respiratory drugs before they get absorbed, with peptide and nucleotide-based drugs being especially vulnerable to metabolism.” 

Reviewer 3 Report

The review " Strategies to enhance drug absorption via nasal and pulmonary routes" by Ghadiri et al., focuses on describing the basic concepts for the transmucosal delivery of drugs, in particular the main strategies for the improvement of pulmonary and nasal delivery by using absorption enhancers. The manuscript is well written and certainly can be of interest for the readers of the Journal, but important and extensive changes would make the article clearer and increase its quality. I suggest the publication of the manuscript in Pharmaceutics after major revision.

Lines 26, 28 and 32 the word “administration” is more appropriate then “delivery” because the authors describe the chemical strategies to enhance systemic drug absorption via nasal and pulmonary routes and not for local nasal or pulmonary delivery by carriers.

Line 73, 75 and 83, the word “systems” should be eliminated because absorption enhancers are surface active molecules not delivery systems, the word “delivery” should be replaced with “administration”.

Line 84, the word “delivery” should be replaced with “administration”.

Il all the manuscript several abbreviations have been reported and used 2 or 3 times. In my opinion there are not necessary and is better to report the complete words to facilitate the text compression.

Line 112, the word “delivery” should be eliminated.

Line 117, the abbreviation GDCA have to be eliminated.

Line 170, in my opinion cyclodextrin, that are able to load the drugs inside their hydrophobic cavity, should be reported in “Other strategies to enhance absorption” like other carrier (e.g. nanoparticles and liposomes).

Line 338, Introduction, add "s" in nanocarrier.

Line 364, the importance of liposomes containing glycerol, so called glycerosomes, for lung administration and related references must be reported.

The authors must take into account that the sentence “nasal or pulmonary delivery” must be used referring to a local accumulation of drug while “nasal or pulmonary administration” can provide local penetration or systemin permeation of drug. In my opinion all these concepts are confused in the review and have to be clarified. Additionally, the expression “delivery systems” must be used for systems (supramolecular structures) able to carry the drugs through the biological barriers. Permeation enhancers like surfactants, phospholipids and polymers are molecules able to chemically modify the drug penetration (local) or permeation (systemic).

Author Response

Point 1- Lines 26, 28 and 32 the word “administration” is more appropriate then “delivery” because the authors describe the chemical strategies to enhance systemic drug absorption via nasal and pulmonary routes and not for local nasal or pulmonary delivery by carriers.

Response-  The sentence has been amended to: “The purpose of this review is a critical assessment of recent advances in the field of absorption enhancers for nasal and pulmonary drug administration.”

Point 2-Line 73, 75 and 83, the word “systems” should be eliminated because absorption enhancers are surface active molecules not delivery systems, the word “delivery” should be replaced with “administration”.

Response- This has been amended as suggested

Point 3- Line 84, the word “delivery” should be replaced with “administration”.

Il all the manuscript several abbreviations have been reported and used 2 or 3 times. In my opinion there are not necessary and is better to report the complete words to facilitate the text compression.

Response- As suggested, unnecessary abbreviation have been removed and changed to the complete words.

Point 4- Line 112, the word “delivery” should be eliminated.

Response- It was amended.

Point 5- Line 117, the abbreviation GDCA have to be eliminated.

Response- It was amended.

Point 6- Line 170, in my opinion cyclodextrin, that are able to load the drugs inside their hydrophobic cavity, should be reported in “Other strategies to enhance absorption” like other carrier (e.g. nanoparticles and liposomes).

Response- As suggested, this part has been moved to “other strategies”.

Point 7- Line 338, Introduction, add "s" in nanocarrier.

Response- It was amended.

Point 8- Line 364, the importance of liposomes containing glycerol, so called glycerosomes, for lung administration and related references must be reported.

Response- Thank you for the suggestion, this section was added to the manuscript.

Glycerosomes are vesicles composed of phospholipids, glycerol, and water, used as novel vesicular carriers for trans-mucosal drug delivery. Glycerosomes are a novel approach that modifies the liposome bilayer fluidity due to its high percent of glycerol, enhancing liposomal properties. Glycerosomes are made from diverse phospholipids and high percent of harmless glycerol (20–40%, v/v). They are flexible vesicular carriers constituted by different substances such as cholesterol that enhance the lipidic bilayer stability. They also may contain basic or acidic lipid molecules which adjust the electrical charge of the vesicular surfaces and decrease liposome aggregation. Glycerosomes can be prepared by the same common techniques used for the preparation of conventional liposomes. For example, curcumin was incorporated in glycerosoms for delivery into the lungs. In this study, curcumin was loaded into glycerosoms which were then combined with two polymers: sodium hyaluronate and trimethyl chitosan to form polymer-glycerosomes [99]. This study showed that nebulized curcumin vesicles were able to protect in vitro A549 cells stressed with hydrogen peroxide, restoring healthy conditions, not only by directly scavenging free radicals but also by indirectly inhibiting the production of cytokine IL6 and IL8. Also, in vivo results in rats showed the high capacity of these glycerosomes to favour the curcumin accumulation in the lungs confirming their potential use as a pulmonary drug delivery system. In another study, Rifampicin loaded glycerosomes, vesicles composed of phospholipids, glycerol and water, were combined with trimethyl chitosan chloride (TMC) to prepare TMC-glycerosomes or, with sodium hyaluronate (HY) to obtain HY-glycerosomes [100]. These new hybrid nanovesicles were tested as carriers for pulmonary delivery. Rifampicin nano-incorporation in vesicles reduced the in vitro drug toxicity on A549 cells, as well as increased its efficacy against Staphylococcus aureus. Finally, the in vivo biodistribution and accumulation, evaluated after intra-tracheal administration to rats, confirmed the improvement of rifampicin accumulation in lung.

Reviewer 4 Report

This review appears rather dated and I am not sure that it would add much to the field. The one potentially interesting section (technologies in development) is remarkably outdated and needs complete rewrite. Writing needs to be far more pedantic and demonstrate more knowledge of the field (some obvious incorrect statements included). Other comments below. 

Line 8 (abstract): replace ‘genes’ with ‘nucleic-acid based’

Line 21: and throughout the manuscript – replace ‘membrane’ with appropriate and non-confusing term such as ‘epithelium’ or ‘mucosal barrier’. Reserve ‘membrane’ when you are referring to cell membranes – otherwise leads to confusion 

Line 34: what is meant by distortion of alveolar epithelial cell layer. Use clearer terminology. 

Line 43: typo paracellular 

Line 62: large and polar drugs can be absorbed by a paracellular mechanism. This is incorrect. Take a look at capacity of this route and revise. 

Figure 2. Depiction of basement membrane unusual – looks like cells. Suggest make it fibrous structure  

Line 64: ‘next barrier is the capillary endothelium’. In figure 2 you have basement membrane so mention it here as well. 

Figure 3. unclear what the blue cell is??

Line 89: no need for ‘E) alkylglycosides’ – these are non-ionic surfactants and can be listed under C)

Line 101: typo enhancers 

Lines 91-110: can you please comment how materials found in the lung are also being used as absorption enhancers? Why don’t they affect lung barrier integrity in native state? Is it a dose-dependent effect. Some mechanistic insight would help – in current form the manuscript is quite superficial. 

Line 162: you may want to comment on epithelial toxicity of AGs - see ‘Epithelial toxicity of alkylglycoside surfactants. J Pharm Sci 2013, 102:114-25.’

Line 174: delete for ‘example’ as the example does not relate to previous statement (related to hydrophilic molecules, not hydrophobic)

Line 180: ‘wildly’? a bit strong?

Line 187: more effective in animal studies compared to what; in vitro?? Clarify

Line 193: accelerate may be inappropriate - it surely is a capacity issue rather than rate

Line 205: ‘enzymes and proteases’. Replace with ‘enzymes, including proteases’

Line 211 and elsewhere: ‘they get absorbed’. Suggest use more formal language 

Lines 205-232: paragraph too long – break it into 2 or more shorter ones – plenty of opportunity to do that 

Line 247: ‘without toxicity towards epithelial cells’ - not usually - this reference could be more of an exception rather than the rule. Consider revising sentence

Line 252 - what is it meant by ‘proper delivery’? writing needs to be more pedantic and precise

Lines 245-292 – huge paragraph; break it into 3-4 smaller ones 

Line 260 – Calu-3 

Line 261: hypnotised – hypothesised?

Line 265-266 – sentence poorly written – rewrite 

Lines 272-275: do dimers and trimers have better solubility? not to my knowledge. please provide reference

Line 281: add w/w or w/v etc with 0.1% 

Line 290: not sure what 'pharmacological availability' is. either say bioavailability or pharmacological effect

Line 291: mm? do you mean micrometer?

Line 304: ‘significantly increased’…compared to what? please write clearly and fully otherwise text does not make sense

Line 308: correct ‘low molecular drugs’

Line 309: peptides and proteins…mainly absorbed through the paracellular route – not really! This is worrying lack of appreciation of the paracellular route anatomy and absorption capacity. Please rewrite

Line 311: potent as what? stronger at doing what? information is incomplete – please add

Line 234: claudin-4 modulator – repetition with above text 

Line 330: which epithelial layer?

Line 334: reference 20 – it is stated that this study used Caco-2 which are intestinal - re-check the study if they used these cells

Line 341: replace ‘to overcome mucosal clearance mechanisms’ with ‘to develop drug delivery strategies to overcome..’

Line 342 delete minus sign (-5)

Section 4.1 has nothing on nanoparticles. The section is on mucus instead! Either add a good chunk of text on nanoparticles or delete it

Lines 382-382 – rewrite 

Lines 393-394: administration of what? exosomes? revise sentence as unclear

Line 396: rhodamine 

Line 397: ‘drug delivery purpose’ - what route? airway / nasal? If not, is this relevant?

Section 4.4: either make it relevant (i.e. airway/nasal administration) or remove it altogether

Section 4.5: can be merged with NP section if you decide to retain the latter 

Line 417: unclear what is meant by 'disturbance'

Table 2: table remarkably outdated, with almost all listed information redundant - companies and technologies from decades ago that are obsolete (e.g. Archimedes, Critical PharmaCEUTICALS, Delsite Biotech and their technologies are no longer in existence). Needs complete rewrite. And as this section is probably the most interesting in the entire paper, I would suggest that authors spend a good deal of time carefully researching CURRENT technologies and summarizing them in a comprehensive table, with references included.  

Conclusion needs complete rewrite as it is vague and lines 445-447 suggest there are current products on the market

Author Response

Point 1- Line 8 (abstract): replace ‘genes’ with ‘nucleic-acid based’

Response- It was replaced.

Point 2- Line 21: and throughout the manuscript – replace ‘membrane’ with appropriate and non-confusing term such as ‘epithelium’ or ‘mucosal barrier’. Reserve ‘membrane’ when you are referring to cell membranes – otherwise leads to confusion

Response- This part was amended to make it less confusing

Point 3- Line 34: what is meant by distortion of alveolar epithelial cell layer. Use clearer terminology.

Response- It was replaced with alteration of the epithelial cell membrane.

 Point 4- Line 43: typo paracellular

Response- It was amended.

 Point 5- Line 62: large and polar drugs can be absorbed by a paracellular mechanism. This is incorrect. Take a look at capacity of this route and revise.

Response- It was amended to “The paracellular transport is not suitable for the transport of large macromolecules and is generally restricted to the compounds of molecular radii less than 11Å. Hydrophilic drugs with low molecular weight, peptides and proteins, have often poor bioavailability. However, it has been shown that some peptide drugs, such as octreotide, desmopressin, and thyrotropin-releasing hormone can be absorbed by this route.”

Point 6- Figure 2. Depiction of basement membrane unusual – looks like cells. Suggest make it fibrous structure 

Response- It was amended.

Point 7- Line 64: ‘next barrier is the capillary endothelium’. In figure 2 you have basement membrane so mention it here as well.

Response- It was added to the text.

Point 8- Figure 3. unclear what the blue cell is??

Response- The blue colour has been changed to neutral, representing a goblet cell, producing mucus.

Point 9- Line 89: no need for ‘E) alkylglycosides’ – these are non-ionic surfactants and can be listed under C)

Response- This section was amended and more examples added.  e) Bio-surfactants, surfactants produced by bacteria, are another type of surfactants investigated as drug absorption enhancers [54]. One of the most well characterized classes of bio-surfactants is rhamnolipids. The effect of rhamnolipids on the epithelial permeability of fluorescein isothiocyanate-labelled dextrans 4 kDa and 10 kDa (named FD-4 and FD-10, respectively), across Caco-2 and Calu-3 monolayers was tested [55]. It was shown that rhamnolipids increased permeation of FD-4 and FD-10 across both cell lines at a safe concentration with a dose-dependent effect. Other type of bio-surfactans are animal derived surfactants. The animal-derived surfactant poractant alfa (Curosurf®) was used to deliver polymyxin E and gentamicin to the lung in a neonatal rabbit model [56]. In this study polymyxin E was mixed with poractant alfa and administered to the near-term rabbit model. This mixture increased bactericidal effect of the antibiotics against Pseudomonas aeruginosa in vivo. This may be due to a more efficient spreading mediated by interactions between drugs and surfactant.

Point 10- Line 101: typo enhancers

Response- It was amended.

Point 11- Lines 91-110: can you please comment how materials found in the lung are also being used as absorption enhancers? Why don’t they affect lung barrier integrity in native state? Is it a dose-dependent effect? Some mechanistic insight would help – in current form the manuscript is quite superficial.

Response- WE belive there has been a misunderstanding as bio-surfactant are not human -derived.  The following sentemnce has been added to increase clarity:

‘Bio-surfactants are surface-active substances synthesised by living cells such as bacteria, fungi and yeast. Biosurfactants are generally non-toxic, environmentally benign and biodegradable’.

Point 12- Line 162: you may want to comment on epithelial toxicity of AGs - see ‘Epithelial toxicity of alkylglycoside surfactants. J Pharm Sci 2013, 102:114-25.’

Response- Thanks for the suggestion. The reference was added and explanation included in the text.

Point 13- Line 174: delete for ‘example’ as the example does not relate to previous statement (related to hydrophilic molecules, not hydrophobic)

Response- It was amended

Point 14- Line 180: ‘wildly’? a bit strong?

 Response- It was amended to broadly.

Point 4- Line 187: more effective in animal studies compared to what; in vitro?? Clarify

Response- This has been amended for clarity

 Point 4- Line 193: accelerate may be inappropriate - it surely is a capacity issue rather than rate

Response- Amended

Point 4- Line 205: ‘enzymes and proteases’. Replace with ‘enzymes, including proteases’

Response- Amended as suggested

Point 4- Line 211 and elsewhere: ‘they get absorbed’. Suggest use more formal language

Response- Amended

Point 4- Lines 205-232: paragraph too long – break it into 2 or more shorter ones – plenty of opportunity to do that

Response- Amended as per suggestion.

Point 4- Line 247: ‘without toxicity towards epithelial cells’ - not usually - this reference could be more of an exception rather than the rule. Consider revising sentence

Response- Amended as per suggestion.

Point 4- Line 252 - what is it meant by ‘proper delivery’? writing needs to be more pedantic and precise

Response- Amended.

 Point 4- Lines 245-292 – huge paragraph; break it into 3-4 smaller ones

Response- Amended as suggested

 Point 4- Line 260 – Calu-3

Response- corrected.

 Point 4- Line 261: hypnotised – hypothesised?

Response- corrected.

 Point 4- Line 265-266 – sentence poorly written – rewrite

Response- sentence was rephrased to “Chitosan and its derivatives,  are excellent examples of cationic polyelectrolyte, they have been extensively used to develop mucoadhesive polymers.”

Point 4- Lines 272-275: do dimers and trimers have better solubility? not to my knowledge. please provide reference

Response- Reference was added to the text.

 Point 4- Line 281: add w/w or w/v etc with 0.1%

Response- W/W added.

Point 4- Line 290: not sure what 'pharmacological availability' is. either say bioavailability or pharmacological effect

Response- It was changed to ‘pharmacological effect’.

Point 4- Line 291: mm? do you mean micrometer?

Response- It was corrected.

Point 4- Line 304: ‘significantly increased’…compared to what? please write clearly and fully otherwise text does not make sense

Response- It was corrected to “Sperminated dextrans increased pulmonary absorption of insulin in rats and permeation of FD-4 through Calu-3 cell monolayers [79].

Point 4- Line 308: correct ‘low molecular drugs’

Response- It was corrected “Hydrophilic drugs with low molecular weight’

Point 4- Line 309: peptides and proteins…mainly absorbed through the paracellular route – not really! This is worrying lack of appreciation of the paracellular route anatomy and absorption capacity. Please rewrite

Response- Apologies- the sentence has now been amended to:

‘The paracellular transport is not suitable for the transport of large macromolecules and is generally restricted to the compounds of molecular radii less than 11Å. Hydrophilic drugs with low molecular weight, peptides and proteins, have often poor bioavailability. However, it has been shown some peptide drugs, such as octreotide, desmopressin, and thyrotropin-releasing hormone absorbed by this route’.

Line 311: potent as what? stronger at doing what? information is incomplete – please add

Response- It was corrected to “extremely potent in opening these tight junctions”

 Line 234: claudin-4 modulator – repetition with above text

Response- It was corrected.

 Line 330: which epithelial layer?

Response- It was changed to “AT-1002, a hexamer peptide, induced tight junction disassembly in the epithelial layer in the trachea [27]. In this study, in vivo intratracheal administration of salmon calcitonin in rats with 1 mg of AT-1002 resulted in a 5.2-fold increase in absorption of calcitonin over the control group [27].”

 Line 334: reference 20 – it is stated that this study used Caco-2 which are intestinal - re-check the study if they used these cells

Response- It was revised.

Line 341: replace ‘to overcome mucosal clearance mechanisms’ with ‘to develop drug delivery strategies to overcome..’

Response- It was revised.

 Line 342 delete minus sign (-5)

Response- It was corrected.

Section 4.1 has nothing on nanoparticles. The section is on mucus instead! Either add a good chunk of text on nanoparticles or delete it

Response- It was revised and this section was added:

 Nanoparticles (NPs) have also been used as a drug carriers for overcoming mucociliary clearance and alveolar macrophages, hence increase the absorption of drugs via respiratory system. Studies showed that NPs can deposit at the lining fluid and escape both mucociliary clearance and alveolar macrophage.  Alveolar macrophages are less efficient to phagocytose ultrafine particles compared to larger particles. NPs also overcome the mucus barrier and achieve longer retention time at the cell surface, effectively penetrating the mucus layer and accumulating on the epithelial surface. In order to penetrate the mucus, nanoparticles must avoid adhesion to mucin fibres and be small enough to avoid significant steric inhibition by the dense fibre mesh. In a study by Schneider, variously sized, polystyrene-based mucus penetrating particles (MPP) were synthesized and their absorption in the lungs following inhalation investigated. They demonstrated MPP as large as 300 nm exhibited uniform distribution and markedly enhanced retention in the lung.

 Lines 382-382 – rewrite

Response- It was rephrased to “Exosomes have been developed as drug delivery vehicles for a variety of drugs.”

Lines 393-394: administration of what? exosomes? revise sentence as unclear

Response- It was rephrased to “After intranasal administration of exosomes loaded with potent antioxidant, catalase, and considerable amount of catalase, was detected in Parkinson’s disease (PD) mouse brain model.”

Line 396: rhodamine

 Response- It was corrected.

Line 397: ‘drug delivery purpose’ - what route? airway / nasal? If not, is this relevant?

Response- It was revised to nasal.

Section 4.4: either make it relevant (i.e. airway/nasal administration) or remove it altogether

Response- It was revised to include relevant nasal/pulmonary studies using CPPs.

 Section 4.5: can be merged with NP section if you decide to retain the latter

Response: The structure of this section has been amended. The mucus related section has now been moved to the mucus barrier section  and added specific NPs for nasal/pulmonary drug absorption to this part.

Line 417: unclear what is meant by 'disturbance'

Response- Sentence was rephrased.

 Table 2: table remarkably outdated, with almost all listed information redundant - companies and technologies from decades ago that are obsolete (e.g. Archimedes, Critical PharmaCEUTICALS, Delsite Biotech and their technologies are no longer in existence). Needs complete rewrite. And as this section is probably the most interesting in the entire paper, I would suggest that authors spend a good deal of time carefully researching CURRENT technologies and summarizing them in a comprehensive table, with references included. 

Response- Thnak you for the suggestions- since the above mentioned companies have sold their technology to other pharmaceutical companies, I have now updated Table 2 with new companies and products under development.

 Conclusion needs complete rewrite as it is vague and lines 445-447 suggest there are current products on the market

Response- it was rephrased.

Round  2

Reviewer 1 Report

N/A

Author Response

Please check the references especially 105:

It is corrected.

Reviewer 3 Report

Please check the references especially 105

Author Response

Point 1- Please check the references especially 105

Response- It was corrected to: 

Lu J, Li N, Gao Y, et al. The Effect of Absorption-Enhancement and the Mechanism of the PAMAM Dendrimer on Poorly Absorbable Drugs. Molecules2018;23(8):2001.  doi:10.3390/molecules23082001

Reviewer 4 Report

The authors have addressed most of the comments previously raised predominantly in a satisfactory manner. However, there are a number of language/grammar errors in the new text. This is rather frustrating and I am not going to highlight all issues here, but careful final check of writing is essential prior to submission. 

Table 2 requires additional improvement since the column heading 'biological product' is inappropriate given that the table lists multiple small drug molecules.  

Author Response

The authors have addressed most of the comments previously raised predominantly in a satisfactory manner. However, there are a number of language/grammar errors in the new text. This is rather frustrating and I am not going to highlight all issues here, but careful final check of writing is essential prior to submission.

The manuscript was grammatically checked by Professor Young.

Table 2 requires additional improvement since the column heading 'biological product' is inappropriate given that the table lists multiple small drug molecules.

 It was amended.